# Expression of the Pro-Fibrotic Marker Periostin in a Mouse Model of Duchenne Muscular Dystrophy

**DOI:** 10.3390/biomedicines12010216

**Published:** 2024-01-18

**Authors:** Jessica Trundle, Viktorija Cernisova, Alexis Boulinguiez, Ngoc Lu-Nguyen, Alberto Malerba, Linda Popplewell

**Affiliations:** 1Department of Biological Sciences, School of Life Sciences and Environment, Royal Holloway University of London, Egham TW20 0EX, UK; j.trundle@ucl.ac.uk (J.T.); viktorija.cernisova.2013@live.rhul.ac.uk (V.C.); alexis.boulinguiez@rhul.ac.uk (A.B.); ngoc.lu-nguyen@rhul.ac.uk (N.L.-N.); l.popplewell@tees.ac.uk (L.P.); 2Developmental Biology and Cancer Research and Teaching Department, University College London Great Ormond Street Institute of Child Health, London WC1N 1EH, UK; 3National Horizons Centre, Teesside University, Darlington DL1 1HG, UK

**Keywords:** Duchenne muscular dystrophy, muscle pathology, periostin, fibrosis

## Abstract

Duchenne muscular dystrophy (DMD) is characterised by fibrotic tissue deposition in skeletal muscle. We assessed the role of periostin in fibrosis using *mdx* mice, an established DMD murine model, for which we conducted a thorough examination of periostin expression over a year. RNA and protein levels in diaphragm (DIA) muscles were assessed and complemented by a detailed histological analysis at 5 months of age. In dystrophic DIAs, periostin (*Postn) mRNA* expression significantly exceeded that seen in wildtype controls at all timepoints analysed, with the highest expression at 5 months of age (*p* < 0.05). We found *Postn* to be more consistently highly expressed at the earlier timepoints compared to established markers of fibrosis like transforming growth factor-beta 1 (*Tgf-β1*) and connective tissue growth factor (*Ctgf*). Immunohistochemistry confirmed a significantly higher periostin protein expression in 5-month-old *mdx* mice compared to age-matched healthy controls (*p* < 0.01), coinciding with a significant fibrotic area percentage (*p* < 0.0001). RT-qPCR also indicated an elevated expression of *Tgf-β1*, *Col1α1* (collagen type 1 alpha 1) and *Ctgf* in *mdx* DIAs compared to wild type controls (*p* < 0.05) at 8- and 12-month timepoints. Accordingly, immunoblot quantification demonstrated elevated periostin (3, 5 and 8 months, *p* < 0.01) and Tgf-β1 (8 and 12 months, *p* < 0.001) proteins in the *mdx* muscle. These findings collectively suggest that periostin expression is a valuable marker of fibrosis in this relevant model of DMD. They also suggest periostin as a potential contributor to fibrosis development, with an early onset of expression, thereby offering the potential for timely therapeutic intervention and its use as a biomarker in muscular dystrophies.

## 1. Introduction

Duchenne muscular dystrophy (DMD) is a rare neuromuscular disease affecting 1:3500–1:5000 newborn boys worldwide, with a worldwide prevalence estimate from 10.71 to 27.78 per 100,000 [1,2]. The disease is characterised by mutations in the DMD gene leading to a lack of dystrophin, a protein crucial for myofibre structure and stability [3]. The pathological progression of DMD involves chronic inflammation, muscle fibre degeneration and subsequent regeneration, leading to the activation of fibrogenic pathways and the deposition of extracellular matrix (ECM) components, ultimately culminating in fibrosis [4]. Fibrosis is characterised by excessive collagen deposition and tissue remodelling, which significantly contribute to the functional decline of affected muscles [4,5].

Whilst muscular dystrophy management has advanced, improving life expectancy [6], in the context of DMD-associated fibrosis, the current clinical therapeutic approach is that of symptom management, with angiotensin-converting enzyme inhibitors showing promise in mitigating cardiac fibrosis [7]. Investigational therapies showing the mitigation of fibrosis have also included glutathione precursor N-acetylcysteine, which improved 8-week-old *mdx* DIA function and fibrotic pathology [8], and the transient receptor potential cation channel, subfamily V, member 2 (TRPV2) stretch-sensitive calcium channels inhibition, which acts by inhibiting the Tgf-β1/TAK1 signalling pathway [9,10]. L-Arginine treatment was also shown to increase nitric oxide production and significantly reduce eccentric contraction-induced damage [11] while nitric oxide production via the phosphodiesterase inhibitor Sildenafil did not demonstrate pathological improvement in two phase 3 trials [12,13].

In the context of DMD-associated fibrosis, a pivotal cellular player is the fibro/adipogenic progenitor (FAP) cell population. FAPs are mesenchymal progenitor cells residing in the skeletal muscle interstitium, capable of differentiating into fibroblasts and adipocytes. The dysregulated activation and persistence of FAPs contribute to the pro-fibrotic microenvironment observed in DMD muscles [14]. Several growth factors are also involved in fibrosis deposition in dystrophic muscles, mediating the ECM build up through FAP differentiation and fibroblast activation. Therefore, they have been explored as potential biomarkers or therapeutic targets for antifibrotic effect: Transforming growth factor-beta 1 (Tgf-β1), a major pro-fibrotic factor, is present at high levels in both the muscles of DMD patients [15] and muscles of the *mdx* mouse model of the disease, such as in the diaphragm (DIA) [16,17]. It has been shown that periostin, another growth factor, promotes Tgf-β1 synthesis as well as activates Tgf-β1 signalling pathways in vascular smooth muscle, the myocardium and the epithelium [18,19,20], and so to enhances Tgf-β1-induced fibrosis in kidneys [21]. Periostin is overexpressed in a variety of fibrotic disorders, where it acts by interacting with bone morphogenetic protein 1 (BMP-1) and promotes collagen fibrillogenesis through collagen cross-linking via the proteolytic activation of lysyl oxidase [22].

In skeletal muscle, periostin is also strongly upregulated in response to injury [23,24]. Conditions such as muscular dystrophy, result in a state of chronic injury leading to skeletal muscle fibrosis [5]. The contribution of periostin to this process is supported by the finding that an increased expression of periostin in the δ-sarcoglycan null mouse model of muscular dystrophy (δ-sarcoglycan null, (Sgcd(−/−)) exacerbates disease pathology, whereas its deletion has a positive effect on the reduction of fibrosis [25]. For DMD, periostin has been reported to be upregulated in the gastrocnemius muscles of 6-week- and 3-month-old *mdx* mice [26], and the DIAs of 3-month-old *mdx*-4cv mice [27]. Moreover, the *mdx* DIA exhibits the most pronounced fibrotic levels across different age cohorts; however, temporal changes in periostin were not assessed [28]. Consequently, we have focused our study on this specific muscle.

Here, we have analysed the expression of periostin at the RNA and protein level in the DIA muscle from *mdx* mice and wildtype control mice, examining how the expression changes over time. Our study reveals that periostin exhibits elevated expression levels in the *mdx* mouse model compared to those in healthy mice. Notably, this heightened expression is observed at all measured age points. Furthermore, this observation aligns with the concurrent increase in fibrosis, as evidenced by increased collagen deposition in the 5-month-old DIA from this DMD animal model.

Understanding the contribution of periostin to skeletal muscle injury and fibrosis holds substantial potential in unveiling novel avenues for therapeutic intervention and disease biomarker identification within the context of the muscular dystrophy phenotype.

## 2. Materials and Methods

All animal procedures were performed in accordance with UK government regulations and were approved by the UK Home Office under Project License P36A9994E. Ethical and operational permission for in vivo experiments was granted by the Animal Welfare Committee of Royal Holloway University of London. *Mdx* (C57BL/1-ScSn-Dmd*mdx*) and C57/BL10 mice were maintained in a standard 12-h light/dark cycle with free access to food and water. Samples were collected according to the TREAT-NMD protocol DMD_M.1.2.007. Male *mdx* and C57/BL10 mice were used for this study at ages of 3, 5, 8 and 12 months, with 5 mice per age group.

Mice were euthanised according to schedule 1 procedures. For each mouse, the diaphragm (DIA) was collected. Half of each DIA was rolled and mounted in optimal cutting temperature (O.C.T.) medium (VWR, Lutterworth, UK) and frozen in isopentane (Sigma, Dorset, UK) chilled in liquid nitrogen. The remaining tissues were snap-frozen in liquid nitrogen for protein and RNA extraction (as described below). All samples were stored at −80 °C. Frozen DIA samples embedded in O.C.T. medium were cryosectioned at a thickness of 10 μm, at 100 μm intervals, using an OTF5000 cryostat (Bright, London, UK). The sections were mounted onto SuperFrost slides (VWR, Leicestershire, UK).

RNA extraction was performed using the RNeasy fibrous tissue mini kit (QIAGEN, Manchester, UK) following the protocol set out by the manufacturer. The concentration of RNA was determined using the Nanodrop ND-1000 spectrophotometer. The OD260/280 ratio given by the spectrophotometer assessed the purity of nucleic acids, with pure nucleic acids having an OD260/280 ratio between 1.8 and 2.0.

cDNA synthesis was performed using the QuantiTect Reverse Transcription Kit (QIAGEN, Manchester, UK). Then, 1000 ng RNA was mixed with 2 µL genomic DNA wipeout buffer and made up to 14 µL with water. This was incubated for 2 min at 42 °C. This was then added to 4 µL of 5× Quantiscript RT buffer, 1 µL RT primer mix, and 1 µL Quantiscript Reverse Transcriptase, to yield a final volume of 20 µL per reaction. This reaction mix was then incubated for 15 min at 42 °C followed by 3 min at 95 °C.

Quantitative polymerase chain reaction (qPCR) was performed using the LightCycler480 system (Roche, Welwyn Garden City, UK) using 384-well plates. Syber green master mix with respective primer pairs was added to the cDNA, yielding a total of 10 μL per sample, which were analysed in triplet. Data were analysed using the LightCycler480 software (Roche, Welwyn Garden City, UK) with the following conditions: activation at 95 °C (5 min), 45 cycles of 95 °C, 57 °C then 72 °C (15 s each) and a melt curve of 95 °C (5 s), 65 °C (60 s) and 97 °C (end). For the relative quantification of the gene of interest, samples were normalised to the mRNA levels of ribosomal protein, large, P0 (*Rplp0*). Primers were purchased from IDT (Leuven, Belgium): *Rplp0* (For:- TTATAACCCTGAAGTGCTCGA, Rev:- CGCTTGTACCCATTGATGATG), *Tgf-β1* (For:- CTGCTGACCCCCACTGATAC, Rev:- GCCCTGTATTCCGTCTCCTT), *Col1α1* (For:- GAAACTTTGCTTCCCAGATGTC, Rev:- AGACCACGAGGACCAGAA), *Postn* (For:- CCTGTAAGAACTGGTATCAAGGT, Rev:- CCTTTCATCCCTTCCATTCTCA), and *Ctgf* (For:- GTGCACTGCCAAAGATGGTG, Rev:- CTTTGGAAGGACTCACCGCT).

Protein extraction from tissue used a 3 mm Tungsten carbide bead (QIAGEN, Manchester, UK) placed in an Eppendorf containing 30 mg of tissue sample in 300 µL RLT lysis buffer, and homogenised using TissueLyser II (QIAGEN, Manchester, UK) at 25 Hz for 4 min with tube rotation after 2 min. Samples were centrifuged at 14,000× *g* for 10 min at 4 °C. The supernatant (protein extract) was decanted to a fresh pre-chilled Eppendorf, and the pellet was discarded. The total protein content of a sample was measured using the DC Protein Assay (BIO-RAD, Watford, UK) following the manufacturer’s standard protocol.

For a Western blot, 10 to 25 µg total protein were mixed with 2 µL reducing agent (Li-Cor, Lincoln, NE, USA), 5 µL of 4× lithium dodecyl sulphate (Li-Cor, Lincoln, NE, USA) and made up to 20 µL with water. Samples were denatured at 70 °C for 10 min. MOPS running buffer (Li-Cor, Lincoln, NE, USA), 4–12% Bis-Tris gels (Li-Cor, Lincoln, NE, USA), and 0.45 µm nitrocellulose membrane (Amersham, GE healthcare, Munich, Germany) were used. For protein detection, goat anti-periostin/OSF-2 Isoform 2 antibody (AF2955; R&D Biosystems, Minneapolis, MN, USA) at 1:2000, rabbit anti-Tgf-β1 antibody (Abcam, Cambridge, UK, ab92486) at 1:500 and the housekeeping mouse anti-vinculin antibody (Sigma, Dorset, UK, SAB4200080) at 1:5000 were used. Secondary antibodies were obtained from Li-Cor (Lincoln, NE, USA). Membranes were visualised using the Odyssey CLx system (Li-Cor, Lincoln, NE, USA) and analysed using the system software Image Studio, version 5.2.

Histological analysis was conducted on 5-month-old *mdx* and C57BL/10 wildtype sections. For immunofluorescence, air-dried slides were rehydrated twice for 5 min in ice-cold PBS before fixation using cold acetone for 10 min at 4 °C. After a quick PBS rinse, blocking solution (2% bovine serum albumin, 5% goat serum, 0.1% triton X-100 in 1× PBS) was added for 1 h at room temperature. Then, sections were incubated with the following primary antibodies: rabbit anti-periostin (Abcam, Cambridge, UK, ab14041, 1:200) and rat anti-laminin (Sigma, Dorset, UK, L0663, 1:1000), diluted in the blocking buffer, overnight, at 4 °C. After 3 rinses of 5 min with PBST (0.05% Tween-20 in 1× PBS), muscle sections were incubated with secondary antibodies: goat anti-rabbit Alexa 568 (Invitrogen, Paisley, UK, 1:200) and goat anti-rat Alexa 488 (Invitrogen, Paisley, UK, 1:200) in PBST goat serum 1% for 1 h at RT. Then, 3 rinses of 5 min with PBST were performed before incubation with DAPI 1:1000 in PBS for 10 min. Finally, after 3 new rinses of 3 min with PBS, the slides were mounted in Mowiol + p-phenylenediamine solution (9 + 1) (Sigma, Dorset, UK) and dried overnight at room temperature. Muscle sections were observed using a Nikon NiE Upright epifluorescence microscope at 20× the objective. Quantifications of staining were performed using Fiji/ImageJ software, version 1.53c.

For Sirius red staining, air-dried slides were fixed using 4% formaldehyde and washed with water and then 100% EtOH. Slides were left alone until completely dried to avoid cytoplasmic staining, then transferred to 0.3% Sirius red solution (aqueous solution in saturated picric acid) for 1 h. The staining was fixed with 2 washes of 0.5% acetic acid and dehydrated in 3 washes of 100% EtOH. Two Xylene baths were used to clarify, and the slides were mounted using DPX mounting solution. Reagents were purchased from Sigma, Dorset, UK. Fibrotic quantification was achieved using ImageJ software, version 1.53c [29]. The auto threshold was applied, adjusting to ensure all fibrotic stained tissue was highlighted. The software then analysed the image for the total area of highlighted fibrotic staining. The fibrotic area was subsequently calculated as the percentages of the total area. The values were recorded for all samples and repeats.

Statistical analysis was performed using the Student’s *t* test to compare wildtype and *mdx* groups as well as different timepoints, using the GraphPad Prism software, version 8, with *p* values < 0.05 considered statistically significant (* = *p* < 0.05, ** = *p* < 0.01, *** = *p* < 0.001, **** = *p* < 0.0001). Any outlier identified using the ROUT method Q = 1%, even if removed, is mentioned in the relevant figure legend.

## 3. Results

### 3.1. Fibrotic-Related Genes Are Differentially Expressed during Disease Progression in mdx Mice Compared to C57BL/10 Mouse Diaphragm Muscles

To understand how periostin expression changes with mice aging, RT-qPCR analysis was performed on RNA harvested from the diaphragms (DIA) of male C57BL/10 and *mdx* mice at 3, 5, 8 and 12 months of age (*n* = 5/group). The levels of the extracellular matrix (ECM) components *Col1α1*, *Postn*, the pro-fibrotic cytokine *Tgf-β1* and key regulator *Ctgf* were evaluated. The relative expression of each marker was obtained by normalising the values by the housekeeping gene *Rplp0*. The relative *Postn* expression was significantly higher in the *mdx* across all age groups relative to the wildtype levels (Figure 1a) ranging from a 9- at 8 and 12 months to over a 50-fold increase at 5 months (*p* = 0.0459), compared to the levels observed in the C57BL/10 control mice. The *Tgf-β1* mRNA expression was also significantly higher in the *mdx* DIA compared to that of the C57BL/10 DIA at 3 (*p* = 0.0120), 8 (*p* = 0.0424) and 12 (*p* = 0.0436) months of age (Figure 1b). The *Tgf-β1* expression in *mdx* muscle was elevated to a similar level at each of the timepoints analysed. *Ctgf*, in contrast, exhibited a disease progression upregulation in DIA in *mdx* mice. At ages of 8 (*p* = 0.0349) and 12 (*p* = 0.0302) months, the *mdx* expression became significantly higher than that of C57BL/10 mice, reaching almost a 5-fold increase (Figure 1c). Furthermore, the *Col1α1* expression was significantly higher in the *mdx* mouse at all ages compared to that in C57BL/10 mice (3 months: *p* = 0.0204, 5 months: *p* < 0.0001, 8 months: *p* = 0.0079, 12 months: *p* = 0.0321), and exhibited a disease progression downregulation (3 vs. 12 months, *p* = 0.0276) in the *mdx* mice (Figure 1d). Our results suggest that some pro-fibrotic markers have a disease progression increase (*Ctgf* 3 vs. 8 months, *p* = 0.0377, 3 vs. 12 months, *p* = 0.0235), while *Tgf-β1* is upregulated consistently for the first year of age (3 vs. 12 months, *p* = 0.4020).

### 3.2. Effect of Age and Disease Progression on Fibrosis-Related Protein Expression in C57BL/10 and mdx Mouse DIAs

A Western blot analysis was performed to quantify the level of periostin and Tgf-β1 protein expression in C57BL/10 and *mdx* mouse DIAs (*n* = 5/group) (Figure 2a). Tgf-β1 protein was chosen for the Western blot analysis due to its known link to periostin with both up- and downstream pro-fibrotic effects, whilst our mRNA analysis revealed *Ctgf* to express significantly at a later timepoint only and does not seem to corelate with Tgf-B1 or periostin expression. Collagen alpha 1 and 3 proteins were instead analysed via Sirius red staining (Figure 3). Protein was extracted from the DIAs at 3, 5, 8 and 12 months of age. The relative protein expression was determined via semi-quantitative analysis using vinculin as a loading control (Figure 2b). No change was observed in the relative periostin expression in C57BL/10 mouse DIAs with age. There were higher periostin protein expressions in the *mdx* DIAs compared to those of C57BL/10 DIAs at all ages, statistically significant at 3 (*p* = 0.000486), 5 (*p* = 0.000742) and 8 (*p* = 0.000083) months. The Tgf-β1 protein expression was consistently more highly expressed in *mdx* DIAs compared to C57BL/10 DIAs, though only significantly in mice at 8 (*p* = 0.004817) and 12 (*p* = 0.001331) months of age (Figure 2b).

### 3.3. DMD Pathology Increased Histological Protein Expression of Periostin and Collagen Types 1 and 3 in mdx Mouse DIA Muscles

To quantify and localise the expression of periostin in muscle, an immunofluorescence of periostin was carried out, together with Sirius red staining targeting collagen type 1 and 3 protein (Figure 3). Due to the *Postn* expression in *mdx* mice peaking at 5 months at the mRNA level, as well as significant upregulation in periostin protein at this time point, DIA muscle samples from 5-month-old *mdx* and C57BL/10 mice (*n* = 4/group) were randomly selected for the immunostaining and histological analyses. Ten images from the largest muscle section were obtained to provide a comprehensive analysis and to select representative images (Figure 3a,b). Image analysis and fibrotic quantification were achieved using ImageJ software [29].

Increased periostin expression was observed and quantified as the percentage of area of a section showing detectable staining (*p* = 0.0031) (Figure 3c). Fibrotic area quantified as a percentage of area stained by Sirius red in the disease model showed over four times higher levels compared to healthy controls (*p* < 0.0001), confirming that fibrosis is present at this time point and that collagen types 1 and 3 are highly expressed in the *mdx* model compared to wildtype mice at 5 months of age (Figure 3d).

## 4. Discussion

In the present study, we have analysed the expression of periostin protein, a marker of fibrosis in the DIA of an *mdx* mouse, a relevant model of DMD, and compared it to those of wildtype C57BL/10 mice in the context of other associated fibrotic markers. Our investigation confirmed the linkage between DMD and the development of skeletal muscle fibrosis, as exemplified by the upregulation in collagen expression within the *mdx* mouse model. This coincides with the upregulation in periostin, confirming its potential as a fibrotic marker for this muscular dystrophy. It is noteworthy that periostin upregulation has been previously documented in other murine models of muscular dystrophy, when contrasted against their wildtype counterparts [25,26,27]. However, periostin expression has not been previously examined in the general context of other pro-fibrotic marker expressions like collagen, Ctgf and Tgf-β1.

The *mdx* mouse is the most commonly used mouse model for DMD, with the DIA being the most severely affected muscle [30,31]. The regeneration of limb muscles has been shown to restore normal muscle structure, whereas the DIA is subject to progressive degeneration and fibrosis formation and shows a pathology similar to that of DMD patients [30,31]. However, it is noteworthy that the *mdx* mouse fails to fully replicate the severity of the fibro-fatty pathology seen in human DMD [32].

Periostin upregulation has previously been described in 6-week and 3-month-old *mdx* mouse medial gastrocnemius [26], as well as in Sgcg−/− [25] and *mdx*4-cv [27] mice. In the DIAs of *mdx* mice, we observed similar significant upregulations in *Postn* mRNA (all timepoints) and periostin protein (3, 5 and 8 months) compared to those in the wildtype control. The peak mRNA expression was detected at 5 months of age, with a more than 50-fold increase compared to that of the 5-month-old C57BL/10 controls. Subsequently we conducted immunofluorescence staining at 5 months, where we once again observed a significant increase in periostin expression within the *mdx* muscle, surpassing over a 30-fold increase compared to that of the wildtype control group. Importantly, this increase in periostin expression correlated with a nearly six-fold increase in the percentage of fibrotic area, suggesting a likely involvement of periostin in the process of fibrosis.

Periostin has previously been reported as a downstream effector of Tgf-β1 [33], a well-established driver of fibrosis [34,35,36]. Here, we show an earlier upregulation in periostin compared to Tgf-β1. Although the design of our study prevents a conclusive determination of the sequential order within the fibrotic mechanism, the data suggest the potential that periostin may express an upstream to Tgf-β1 in a dystrophic environment.

Tgf-β1 is known to regulate the synthesis of collagen [37], which has been reported to be increased three-fold in 2-month-old *mdx* DIAs compared to that in wildtype DIAs [38]. In line with this, we observed the expressions of both *Tgf-β1* and *Col1α1* mRNA to be significantly upregulated in the *mdx* DIAs compared to those in C57BL/10 DIAs at 3 months of age. Furthermore, a nearly 30-fold increase in the expression of *Postn* in the DIA was reported compared to that in wildtype DIAs at 2 months of age [38]. At 3 months, we observed a nearly 15-fold increase; at 5 months, a 50-fold increase; at 8 months, a 9-fold increase; and at 12 months, a 12-fold increase. The *Postn* expression therefore remained relatively upregulated for the full 12-month timeframe.

We did not observe a disease-related change in the *Tgf-β1* transcript or Tgf-β1 protein expression in the *mdx* model, with expression levels remaining comparable between age groups. It was previously shown that in *mdx* mice following cardiotoxin injection, the *Tgf-β1* expression was increased prior to that of *Postn* [39], indicating an upstream function. Furthermore, treatment of *mdx* mice with a Tgf-β1-neutralising antibody results in a decrease in periostin protein expression [40]. These findings collectively suggest that the upregulation of Tgf-β1 induces periostin expression. It has elsewhere been reported that Tgf-β1 works downstream of periostin [20], whilst cross-talk between Tgf-β1 and periostin has been seen as an amplifying mechanism for pulmonary fibrosis [41]. Our study cannot discern which growth factor is activated first and which one is activated in response. However, the similarly raised and constant expression of Tgf-β1 and periostin over time suggests a likely link in their expression.

*Ctgf* exhibited a progressive upregulation in *mdx* mice, with significant increases at 8 and 12 months, contrasting the early significance of *Postn* upregulation. In contrast, we observed a downregulation in *Col1α1* expression with disease progression (*p* = 0.0276), a finding that has previously been reported in the case of procollagen 3 mRNA in *mdx* DIAs [42]. A novel stable isotope approach, using deuterium oxide long-term labelling to measure the slow turnover of collagen in skeletal muscle, showed that there was a significant decline in collagen synthesis and an increase in collagen degradation in old mice compared to young mice [43]. The study suggests that this age-related decline in collagen proteostasis may contribute to the loss of muscle mass and function that occurs with aging but is however limited by the assumption of the complete renewal and turnover of collagen. We suggest that the progressive reduction of collagen seen in the present study may be indicative of *mdx* muscle age, rather than DMD disease progression specifically. As DIA muscles of *mdx* mice become fibrotic early in life, it is also possible that a feedback mechanism downregulates the collagen deposition, as aged muscles naturally present a substantial volume occupied by the extracellular matrix and less cellular tissue (either myogenic or non-myogenic), is present.

Whilst our current study focusses on the commonly reported link between Tgf-β1 and periostin, the mechanism of periostin within the fibrotic cascade in DMD muscle pathology remains speculative. We know the Tgf-β1 signalling pathway to be stimulated by angiotensin II (AngII) [44], promoting fibroblast migration and contractility, as well as increased periostin expression [45], and that a blockade of AngII receptors improves regeneration and decreases fibrosis in skeletal muscle [40,46]. This blockade strategy offers promising therapeutic targeting in dystrophin-deficient cardiomyopathy [47] as, in cardiac fibroblasts, AngII was further found to increase periostin expression via Ras/p38 MAPK/CREB and ERK1/2/Tgf-β1 pathways [48]. A novel immunomodulator was then found to alleviate cardiac hypertrophy and fibrosis through the reduction of periostin [49]. Moreover, pulmonary arterial smooth muscle cells pretreated with an inhibitor of ERK1/2 showed significant prevention of induced periostin expression [50]. A similar significant reduction was also seen with reactive oxygen species (ROS) inhibition, which has been linked to Ras GTPase modulation [51] and AngII sensitivity [52] and is elevated in DMD [53]. Whilst elucidating mechanistic conclusions was not the scope for our study, our demonstration of early periostin temporal expression in fibrotic muscle could suggest a role within the AngII-induced ERK1/2 or ROS signalling pathways, upstream of Tgf-β1.

Altogether, we provide further evidence of the involvement of periostin in fibrotic pathology in skeletal muscle using a model of muscular dystrophy. We have shown a positive correlation of *Postn* expression within the *mdx* mouse model for DMD compared to that within the control, which coincided with an increase in fibrotic deposition shown through DIA collagen staining. We therefore highlight the potential of periostin as a contributor of fibrosis. Our data further establish the pro-fibrotic role of periostin in skeletal muscle fibrosis seen in a model of DMD. This has potential implications for the development of improved and more targeted approaches in the treatment of this pathological phenotype of DMD and also other muscular dystrophies.

## Figures and Tables

**Figure 1 biomedicines-12-00216-f001:**
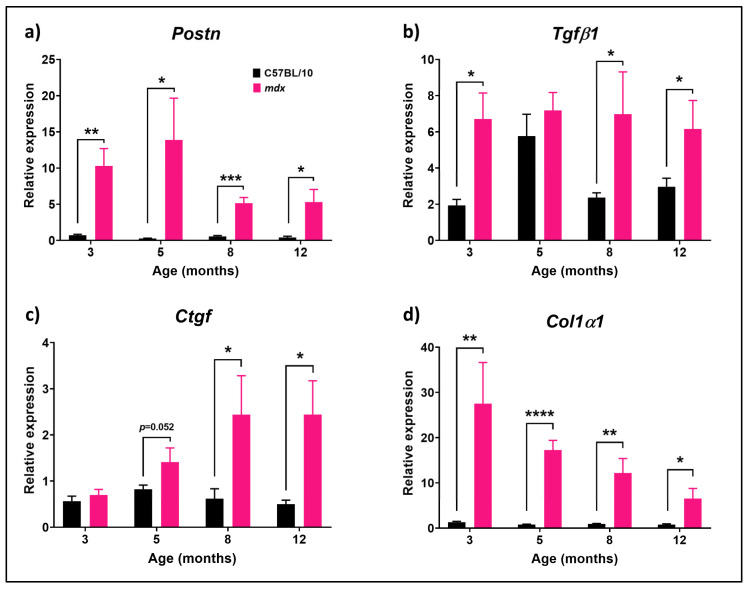
mRNA expression of fibrosis-related genes in the DIA muscle from C57BL/10 and *mdx* mice. RT-qPCRs for (**a**) *Postn*, (**b**) *Tgf-β1*, (**c**) *Ctgf* and (**d**) *Col1α1* were performed on RNA harvested from the DIA muscle of male C57BL/10 and *mdx* mice at 3, 5, 8 and 12 months of age. Relative levels of expression were determined by normalising to the expression level of *Rplp0*. Data shown as means ± S.E.M., *n* = 5 mice per age group and strain. No outliers were identified, and statistical analysis was performed using unpaired Student’s *t* test, *p* < 0.05 (*), *p* < 0.01 (**), *p* < 0.001 (***), *p* < 0.0001 (****) between genotypes and timepoints.

**Figure 2 biomedicines-12-00216-f002:**
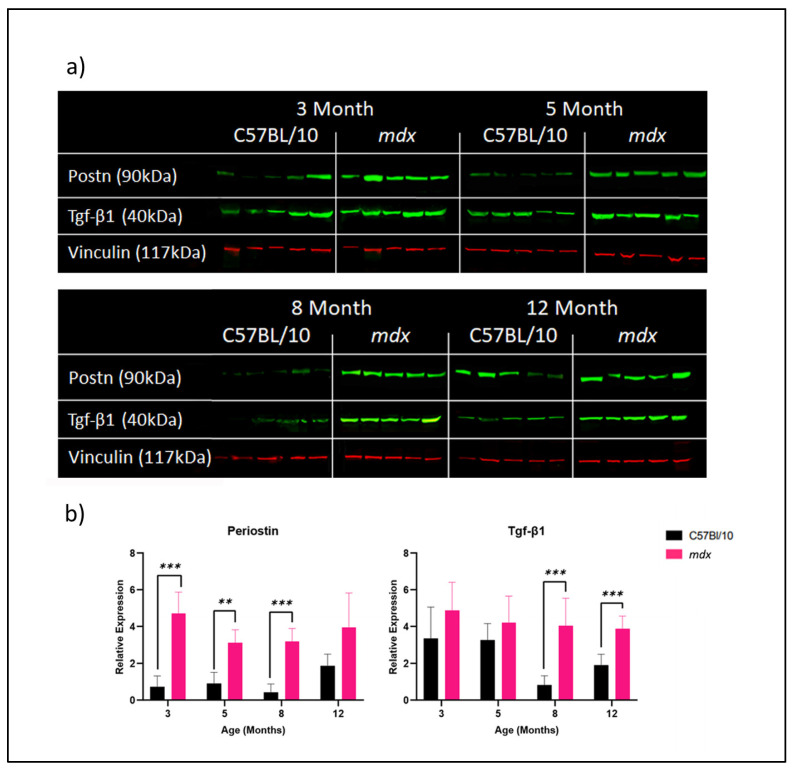
Periostin and Tgf-β1 protein expression in C57Bl/10 and *mdx* mouse DIAs at 3, 5, 8 and 12 months of age. All samples for each age group for each strain were homogenised, and total protein was extracted. In total, 10 µg total protein was separated on a 4–12% Bis-Tris gel followed by a transfer onto a 0.45 µm nitrocellulose membrane. A Chameleon Duo protein ladder (Li-Cor) was used to assess the protein size. The membranes were imaged using the Odyssey CLx system and analysed using ImageStudio software, version 5.2. (**a**) Combined Western blot images of all samples for each age group for periostin, Tgf-β1 and vinculin housekeeping protein expression in the DIAs of C57BL/10 and *mdx* mice at 3, 5, 8 and 12 months of age. (**b**) The relative periostin and Tgf-β1 protein expression was determined via semi-quantitative analysis. The levels of protein were normalised to that of vinculin to determine the relative protein levels. Protein quantification was performed using ImageStudio Software (Li-Cor). Data shown as means ± S.E.M., *n* = 5 mice per age and model unless otherwise stated. Statistical analysis was performed using unpaired Student’s *t* test, *p* < 0.01 (**), *p* < 0.001 (***),and outliers were excluded using the ROUT method: Q = 1%:C57BL/10 Postn 3 months (*n* = 4), Postn 12 months (*n* = 2) and Tgf-β1 8 months (*n* = 4).

**Figure 3 biomedicines-12-00216-f003:**
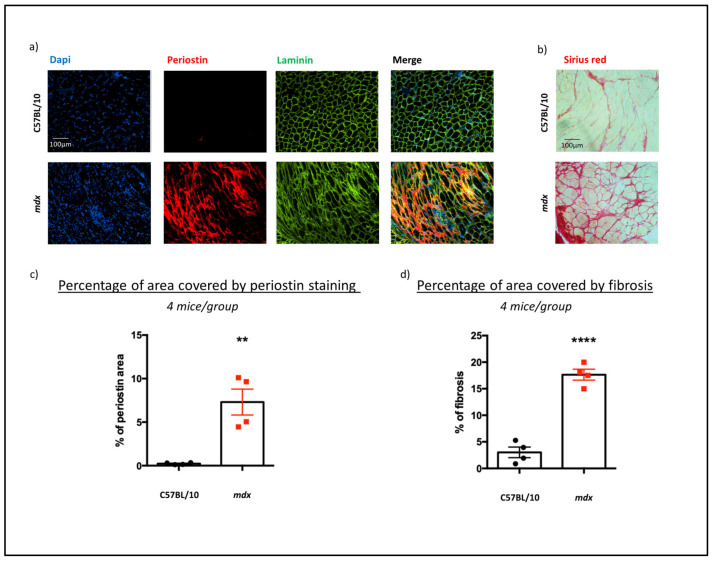
Immunohistological analysis of fibrosis in C57BL/10 and *mdx* mouse DIAs. The levels of muscle fibrosis in the DIAs of 5-month-old C57BL/10 and *mdx* mice were assessed through immunostaining for periostin or Sirius red staining (for collagen 1 and 3). Representative images are shown for (**a**) periostin, with periostin stained in red, laminin in green and DAPI in blue, and (**b**) Sirius red, with collagen 1 and 3 stained in red. The relative quantifications of (**c**) periostin and (**d**) collagen positive areas are shown as the percentages of the total areas of the corresponding images. Data shown as means ± S.E.M., *n* = 4 muscles/mouse model. Statistical analysis was performed using Student’s *t* test, *p* < 0.01 (**), *p* < 0.001 (****).

## Data Availability

The data presented in this study are available upon request from the corresponding author. The data are not publicly available due to further development of the work.

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
