# Peer review of "Expression of the Pro-Fibrotic Marker Periostin in a Mouse Model of Duchenne Muscular Dystrophy"

_biomedicines, 2024, doi:10.3390/biomedicines12010216_

Round 1
Reviewer 1 Report
Comments and Suggestions for Authors
The manuscript adds a small piece in the understanding of the players’ role involved in the fibrotic process.
The Abstract is clear and accurate. It appropriately summarizes the manuscript's main parts.
The manuscript main text is clear and comprehensive.
The introduction is detailed and with adequate bibliographical references.
Materials and methods are also well detailed and describe quite precisely the experiments that the authors performed.
Results are well presented clear and detailed.
The discussion describes the results obtained by the authors and well integrates what is already known.
The only critical issue is not having better explained the conclusions, clarifying how these results can be applied in the discovery/development of new treatments.
The figures/tables are appropriate, and are easy to interpret and understand.
Minor revisions:
1 - Chapter 3 presents an error in the subchapters, the first three are all named 3.1
2 - I suggest to insert the sample size, not only in the figures’ caption, but also in the text.
For example, this data is missing in the chapter: "Fibrotic related genes are differentially expressed during disease progression in mdx 179 compared to C57BL/10 mouse diaphragm muscles". Please align all the text.
3 - minimal error: in the main test the figures are indicated by the number and a capital letters; but in the figures and captions they are indicated by lowercase letter.
Reviewer 2 Report
Comments and Suggestions for Authors
This is a well-designed study using a traditional dystrophin-deficient mouse model. Despite the wealth of information, this pathology continues to reveal many new mechanisms. The authors demonstrated an early periostin response characteristic of the skeletal muscles of mdx mice and, above all, the diaphragm, which is most susceptible to pathological changes in these animals. The results suggest that the periostin level can be used as an early prognostic marker for the development of fibrosis in Duchenne dystrophy, as well as to assess the severity of the pathology. After carefully evaluating the work, I have the following questions.
Major:
The main question concerns the molecular mechanism preceding the increase in periostin levels. What is the reason? The authors should speculate in detail on this topic. In particular, it is known that periostin, like other profibrotic factors, is sensitive to ROS (PMID: 27220372, PMID: 27434868 and others), whose level is significantly increased in Duchenne dystrophy (PMID: 28472288). Moreover, periostin is also known to exhibit calcium ion sensitivity (PMID: 18450759), whose levels are also elevated due to dysregulation of ion homeostasis in dystrophic-deficient muscles (PMID: 36768550). Given the complexity of the pathology, other mechanisms also cannot be excluded. Therefore, I invite the authors to discuss the possible reasons behind the change in periostin levels.
Minor:
It should be clearly stated that mdx mice have relatively mild phenotype compared to wild-type and that they do not recapitulate fibro-fatty progression observed in patients.
Line 128. The sequences of the primers used should be given for the convenience of the reader.
Fig. 2 is practically invisible (especially Western blot images).
Round 2
Reviewer 2 Report
Comments and Suggestions for Authors
The authors adequately responded to my comments.